# Prediction of Antioxidant Capacity of Thiolate–Disulfide Systems Using Species-Specific Basicity Values

**DOI:** 10.3390/antiox13091053

**Published:** 2024-08-29

**Authors:** Tamás Pálla, Béla Noszál, Arash Mirzahosseini

**Affiliations:** 1Department of Pharmaceutical Chemistry, Semmelweis University, 1092 Budapest, Hungary; palla.tamas@semmelweis.hu (T.P.); noszal.bela@semmelweis.hu (B.N.); 2Center for Pharmacology and Drug Research & Development, Semmelweis University, 1085 Budapest, Hungary

**Keywords:** standard redox potential, oxidative stress, redox equilibrium

## Abstract

The principal reactions that maintain redox homeostasis in living systems are the deprotonation of thiols, followed by the oxidative conversion of the produced thiolates into disulfides, which thus reduce the harmful oxidizing agents. The various biological thiols have different molecule-specific propensities to carry on the co-dependent deprotonation and redox processes. This study utilizes the known correlation between thiolate basicities and oxidizabilities, to quantify antioxidant or reducing capacities and pH-dependences of thiol–disulfide antioxidant systems, as a tool to find adequate molecules against oxidative stress.

## 1. Introduction

### 1.1. Basis of Oxidative Stress and Antioxidant Pathways

Reactive oxygen species (ROS) and reactive nitrogen species (RNS) are products of redox metabolism in the body [1,2,3,4]. Their excessive production, however, is associated with serious illnesses, such as cancer, multiple sclerosis, Alzheimer’s, Parkinson’s, and cardiovascular diseases (atherosclerosis, cardiac ischemia, reperfusion), and cystic fibrosis, to name a few [5,6,7,8]. The highly coveted therapy against oxidative stress is currently an unmet medical need.

The prime moieties that protect living systems against oxidative stress are the thiol and selenol groups in various biomolecules. In the protective reactions, thiols undergo deprotonation and subsequent oxidation into disulfides. The thiols have a large-scale propensity of deprotonation and oxidation, depending on the structure and properties of the biomolecule they are part of. The same is true for selenols, although they are a less populous branch of biomolecules.

Understanding, preventing, and curing oxidative stress and its consequences have long been the objective of several efforts. As a part of that, the measurement of antioxidant activities has so far been proposed by nearly two dozen reactions, with no unifiable mechanism at the molecular level. The most important in vitro antioxidant assays, summarized by Gulcin [9], are, for example, oxygen radical absorbance capacity (ORAC), total radical trapping antioxidant parameter (TRAP), Trolox equivalence antioxidant capacity assay (TEAC), and many others. However, of the many antioxidant assays available, only cupric ions (Cu^2+^) reducing antioxidant power assay (CUPRAC) and peroxynitrite scavenging assays are capable of measuring the antioxidant activity of thiolates. The CUPRAC assay, developed by Apak’s group [10], measures antioxidant capacity by reducing Cu^2^⁺ to Cu⁺ in the presence of neocuproine, forming a Cu⁺-neocuproine complex with a peak absorption at 450 nm. This method is cost-effective, stable, and suitable for a wide range of antioxidants, including both hydrophilic and lipophilic types. It operates at a pH close to physiological conditions (7.0), making it more applicable to potential in vivo antioxidant reactions, and is particularly effective for measuring thiol-type antioxidants. Peroxynitrite (ONOO^−^) is a potent and short-lived oxidant that contributes to neurodegeneration and various diseases like Alzheimer’s and cancer. It is formed by the reaction of nitric oxide with superoxide and can diffuse across cell membranes, causing cellular damage through lipid peroxidation, protein oxidation, and DNA strand breakage. Due to its stability relative to other free radicals, peroxynitrite can induce significant tissue injury. Yet there are no endogenous enzymes to inactivate it, highlighting the need for specific scavengers. The oxidation of dihydroxyrhodamine by peroxynitrite can be measured using fluorescence spectrophotometry, providing a method to assess its scavenging activity [11].

### 1.2. Thiol–Disulfide (Selenol–Diselenide) Systems and Their Species-Specific Acid–Base and Redox Equilibria

Cysteine, cysteamine, homocysteine, and glutathione are sulfur-containing molecules with critical biological roles [12]. Cysteine is essential for protein synthesis and acts as a precursor to glutathione, a key antioxidant. Cysteamine serves as a protective agent against radiation and oxidative stress, and is used in treating cystinosis. Homocysteine is an intermediate in methionine metabolism, with elevated levels linked to cardiovascular disease [13]. Glutathione is a powerful antioxidant that protects cells from oxidative damage and detoxifies harmful substances, playing a crucial role in maintaining cellular redox balance [14]. Ovothiol A, B, and C are natural 4-mercaptohistidine derivatives, varying in methylation at the amino site, found in marine organisms, particularly in sea urchin eggs [15]. Ovothiol is a functional group-dense molecule with multiple protonation states, making it one of the most chemically diverse small biomolecules. It is zwitterionic at physiological pH, features strong imidazole-thiolate interactions, and has unique thiolate protonation constants, resulting in its sulfur atom predominantly existing in an anionic form across the pH scale [16]. This versatility enhances its antioxidant capabilities without altering its molecular structure [16]. Ovothiol has a unique ability to scavenge reactive oxygen species (ROS) and protect against DNA, protein, and lipid damage [17].

Thiols can deprotonate to produce thiolates which are the actual species that undergo oxidation to form disulfides [18]. Deprotonation and the subsequent stepwise thiolate–disulfide redox equilibria are depicted in Figure 1, where *K*_r1_ and *K*_r2_ stand for the redox equilibrium constants. *K*_p_ is the protonation constant of the thiolate, while its reciprocal is the acid dissociation constant of the thiol. Formally, the selenium-containing analogs have the same deprotonation and redox equilibria. For brevity, all reaction schemes will be discussed here in terms of sulfur-containing compounds. The interfering and co-dependent acid–base and redox processes can be thoroughly characterized in terms of species-specific parameters only [19]. The detailed description of species-specific (also referred to as microscopic) acid–base and redox equilibrium constants are concisely summarized below [20,21].

Figure 2 depicts the acid–base equilibria of cysteine as an example compound. Macroequilibria (top lines) indicate the stoichiometry of the successively protonating ligand and the stepwise macroscopic *K*_p_ protonation constants. Besides the at least one thiolate basic moiety, the completely unprotonated ligand may have any number of further basic moieties. The basic moieties are symbolized with N, S, O for the cysteine amino, thiolate, and carboxylate groups, respectively. The ligand can bear a maximum of *n* hydrogen ions, *n* being the number of basic moieties, and all the species are also designated lexicographically (a, b, c, …). The microscopic acid–base constants are depicted with lower case *k*, with the superscript indicating the basic moiety undergoing protonation, while the subscript (if any) shows the site(s) already protonated. The relationships between macroscopic and microscopic protonation constants can be derived from their respective definitions pioneered by Niels Bjerrum [22]. Some examples are:(1)$$\beta_{1}=K_{p1}=\frac{\left[{\mathrm{HL}}^{-}\right]}{\left[L^{2-}\right]\cdot \left[H^{+}\right]}$$
(2)$$\beta_{2}=K_{p1}\cdot K_{p2}=\frac{\left[H_{2}L\right]}{\left[L^{2-}\right]\cdot {\left[H^{+}\right]}^{2}}=k^{N}\cdot k_{N}^{S}+k^{S}\cdot k_{S}^{O}+k^{N}\cdot k_{N}^{O}$$
(3)$$\beta_{3}=K_{p1}\cdot K_{p2}\cdot K_{p3}=\frac{\left[H_{3}L^{+}\right]}{\left[L^{2-}\right]\cdot {\left[H^{+}\right]}^{3}}=k^{N}\cdot k_{N}^{S}\cdot k_{\mathrm{NS}}^{O}$$
(4)$$\left[{\mathrm{HL}}^{-}\right]=\left[b\right]+\left[c\right]+\left[d\right],\;\mathrm{therefore}\;K_{p1}=k^{N}+k^{S}+k^{O}$$
where *β*_1_, *β*_2_, and *β*_3_ are the cumulative, macroscopic protonation constants. Depending on the path of protonation, Equations (2) and (3) can be written in 2 and 6 different, respective, equivalent ways with microconstants covering always the path of protonation from completely deprotonated form to the fully protonated form of the ligand.

In order to define standard redox potentials of the two-electron transition reactions between thiolates and disulfides, the appropriate microspecies must be chosen from the acid–base microspeciation scheme (Figure 2). In Figure 3, an example is given with the reactive forms of cysteine and cystine that are most abundant near neutral pH.

The standard redox potential of the redox pair depicted in Figure 3 ($E{{}^{\circ}}_{f^{\prime }/b}$) can be determined by observing the thiolate–disulfide redox equilibrium against a redox pair of known *E*° (e.g., glutathione disulfide/glutathione, GSSG/GSH, which is actually the ‘gold standard’ in thiol–disulfide redox systems, and also the most abundant intracellular pair to maintain redox homeostasis). By observing the apparent redox equilibrium constant (*K*_C_) between cystine/cysteine and GSSG/GSH, one can calculate standard redox potentials as follows:(5)$$E{{}^{\circ}}_{f^{\prime }/b}=E{{}^{\circ}}_{\mathrm{GSSG}/\mathrm{GSH}}+\frac{R\cdot T}{z\cdot F}\cdot \ln k^{f^{\prime }/b}=E{{}^{\circ}}_{\mathrm{GSSG}/\mathrm{GSH}}+\frac{R\cdot T}{z\cdot F}\cdot \ln\left(K_{C}\cdot \frac{\chi_{b}^{2}\cdot \chi_{\mathrm{GSSG}}}{\chi_{f^{\prime }}\cdot \chi_{\mathrm{GSH}}^{2}}\right)$$
where *χ* is the mole fraction of the protonation microspecies in question, and *R*, *T*, *z*, and *F* are the well-known parameters/constants in the Nernst equation. Details of the above method can be found in a recent review [23]. In the current work we aimed to develop the quantification of antioxidant capacity of thiolate- (and selenolate-) containing compounds using their species-specific physico-chemical parameters.

## 2. Materials and Methods

### 2.1. Data Collection

Data for this study were systematically collected from existing literature using a comprehensive search strategy. We utilized multiple academic databases to identify relevant peer-reviewed articles, conference papers, and reviews. The PKAD database [24] was used with the advanced search option to identify data pertaining to cysteine residues. Inclusion criteria were established to select studies that provided quantitative or qualitative data pertinent to our research questions. The collected data were screened independently by two researchers to confirm eligibility.

### 2.2. Statistical Analysis and Mathematical Calculations

The data analysis was conducted using the R programming language (R version 4.0.5—R Foundation for Statistical Computing, Vienna, Austria) and the R Studio integrated development environment (IDE)—https://posit.co/download/rstudio-desktop/, accessed on 22 August 2024—Posit, Boston, MA, USA. Visualization of the results was performed using R Studio and Origin Pro 8 (OriginLab Corp., Northampton, MA, USA) software. Propagation of uncertainty was calculated using the delta method.

## 3. Results

### 3.1. Relationship between Acid–Base and Redox Characteristics

The directly observable, apparent (conditional) thiol–disulfide redox potentials are concentration-weighted resultants of several differently protonating forms, and are therefore pH-dependent values. On the other hand, their constituent, species-specific redox potentials are pH-independent parameters. The standard redox potential values belonging to different microspecies of the same parent compound differ due to the different protonation states of the basic moieties surrounding the thiolate/disulfide group. The standard redox potentials of biorelevant thiolate-containing compounds (mostly amino acids or their derivatives) have recently been shown [19] to be in close relationship with the log*k* values of the same-sulfur thiolate (slope = −0.0605, std err = 0.0011, intercept = 0.1683, std err = 0.0083, adj. R^2^ = 0.9910, *n* = 31, F-value = 3311.3872). The acid–base and redox parameters referenced from work [19] are species-specific ones, characterizing protonation or oxidation transitions on a submolecular level pertaining to a particular protonation state, as opposed to the more frequently used macroscopic p*K*_a_ and apparent redox potential values. Previously reported [25,26,27] macroscopic parameters for cysteine (thiol p*K*_a_ = 8.4, redox potential = −0.22 V) and glutathione (thiol p*K*_a_ = 8.7, redox potential = −0.26 V) are distinct, but in agreement with the species-specific values; i.e., if pH 7 is considered the species-specific acid–base and redox parameters can be calculated to an apparent redox potential in agreement with previous findings. The same is analogously true for selenolate-containing compounds as well, only the linear relationship has an offset of approximately 250 mV [28]. Using this linear regression the standard redox potential of thiolates found in other biological compounds can also be determined using only the thiolate basicity value. A literature search for reported protein thiolate log*k* values in the PKAD database [24] unfortunately affords only a handful of data. These reported values are listed in Table 1 with the calculated standard redox potential values.

The standard redox potentials calculated in Table 1 serve as a demonstration that the thiolate basicity (or conversely thiol p*K*_a_) can be a useful tool in comparing the antioxidant capacity of thiolates (or selenolates) in biologically important molecules, since redox equilibria, especially the site-specific ones are far more complex and difficult to determine, compared to protonation constant values, even though the site-specific protonation constants of proteins (usually ‘group constants’ [29]) are not trivial to determine either. The pH-dependent, apparent (conditional) antioxidant capacity of thiolates is related to the pH-independent, species-specific redox potentials, exemplified by the cysteine (CysSH)—cystine (CysSSCys) system, as follows.

### 3.2. Standard Redox Potential pH Profiles

If a redox electrode is created from a thiolate–disulfide redox pair (e.g., cystine/cysteine), the redox potential of this system can be expressed by the Nernst equation. In terms of cysteine–cystine microspecies, the a′/a, f′/b, k′/d, p′/f, 4 redox pairs exist. The concentration of each of them is in correspondence with the single, overall redox potential of the solution. Once all the redox and acid–base equilibria are taken into account, the redox potential of each redox pair can be calculated using Equation (6) and its analogues:(6)$$E_{a^{\prime }/a}=E{{}^{\circ}}_{a^{\prime }/a}-\frac{R\cdot T}{z\cdot F}\cdot \ln\frac{{\left[a\right]}^{2}}{\left[a^{\prime }\right]}=E{{}^{\circ}}_{\;a\;\prime /a}-\frac{R\cdot T}{z\cdot F}\cdot \ln\frac{{\left({\left[\mathrm{CysSH}\right]}_{T}\cdot \chi_{a}\right)}^{2}}{{\left[\mathrm{CysSSCys}\right]}_{T}\cdot \chi_{a^{\prime }}}$$
where [CysSH]_T_ and [CysSSCys]_T_ are the total concentrations of cysteine and cystine, respectively. As Equation (6) shows $\left[a\right]={\left[\mathrm{CysSH}\right]}_{T}\cdot \chi_{a}$, and $\left[a^{\prime }\right]={\left[\mathrm{CysSSCys}\right]}_{T}\cdot \chi_{a^{\prime }}$, where:(7)$$\chi_{a}=\frac{\left[a\right]}{{\left[\mathrm{CysSH}\right]}_{T}}=\frac{\left[a\right]}{\left[a\right]+\left[b\right]+\left[c\right]+\left[d\right]+\left[e\right]+\left[f\right]+\left[g\right]+\left[h\right]}=\frac{1}{1+K_{p1}\cdot \left[H^{+}\right]+K_{p1}\cdot K_{p2}\cdot {\left[H^{+}\right]}^{2}+K_{p1}\cdot K_{p2}\cdot K_{p3}\cdot {\left[H^{+}\right]}^{3}}$$
(8)$$\chi_{a^{\prime }}=\frac{\left[a^{\prime }\right]}{{\left[\mathrm{CysSSCys}\right]}_{T}}=\frac{\left[a^{\prime }\right]}{\left[a^{\prime }\right]+\left[b^{\prime }\right]+\left[c^{\prime }\right]+\left[d^{\prime }\right]+\left[e^{\prime }\right]+\left[f^{\prime }\right]+\left[g^{\prime }\right]+\left[h^{\prime }\right]+\left[i^{\prime }\right]+\left[j^{\prime }\right]+\left[k^{\prime }\right]+\left[l^{\prime }\right]+\left[m^{\prime }\right]+\left[n^{\prime }\right]+\left[o^{\prime }\right]+\left[p^{\prime }\right]}=\frac{1}{1+K_{p1}^{\prime }\cdot \left[H^{+}\right]+K_{p1}^{\prime }\cdot K_{p2}^{\prime }\cdot {\left[H^{+}\right]}^{2}+K_{p1}^{\prime }\cdot K_{p2}^{\prime }\cdot K_{p3}^{\prime }\cdot {\left[H^{+}\right]}^{3}+K_{p1}^{\prime }\cdot K_{p2}^{\prime }\cdot K_{p3}^{\prime }\cdot K_{p4}^{\prime }\cdot {\left[H^{+}\right]}^{4}}$$

The four redox potentials of the cystine/cysteine redox pairs can be calculated independently at any pH, and their weighed mean and standard deviation can be determined. The weights used for each redox pair are the products of the corresponding χ values (e.g., after Equation (6) the weight of $E_{a^{\prime }/a}$ would be $\chi_{a}\cdot \chi_{a^{\prime }}$, as in Equation (9)). This is to represent the relative abundance of a given redox pair at a certain pH.
(9)$$E_{\mathrm{CysSSCys}/\mathrm{CysSH}}^{\mathrm{app}}=\frac{\chi_{a}\cdot \chi_{a^{\prime }}\cdot E_{a^{\prime }/a}+\chi_{b}\cdot \chi_{f^{\prime }}\cdot E_{f^{\prime }/b}+\chi_{d}\cdot \chi_{k^{\prime }}\cdot E_{k^{\prime }/d}+\chi_{f}\cdot \chi_{p^{\prime }}\cdot E_{p^{\prime }/f}}{\chi_{a}\cdot \chi_{a^{\prime }}+\chi_{b}\cdot \chi_{f^{\prime }}+\chi_{d}\cdot \chi_{k^{\prime }}+\chi_{f}\cdot \chi_{p^{\prime }}}$$

The apparent redox potential pH profiles of some select thiolate–disulfide systems, together with selenocysteine–selenocystine, are depicted in Figure 4. The redox potentials and protonation constants (used to calculate the *χ* values) are taken from [16,19,20,28,30,31,32,33].

### 3.3. Antioxidant Capacities Directly from Thiolate Basicities

The antioxidant capacity of thiolate- and selenolate-containing compounds can also be calculated using the relevant basicity values directly, since it has recently been demonstrated that basicity is in tight correlation with standard redox potential, due to the fact that both physico-chemical parameters are influenced by the nucleophilicity of the sulfur or selenium atom. The advantage of using basicity values directly as a measure of pH-dependent antioxidant capacity is that this parameter is inherently independent of the compound concentrations, since this value is defined for solutions where only the reduced form of the compound is present (although implementation of antioxidant capacity values in application must always take into account the effective concentration of the antioxidant at the site of action). The disadvantage of this parameter is that thiolate- and selenolate-species are no longer comparable, since the log*k*~*E*° relationship of these two classes of antioxidants shows parallel but distinct linear equations, positioned by some 250 mV values apart, i.e., thiolate and selenolate basicities do not translate to the same metric with regards to antioxidant capacity.

The calculation method for this new measure of antioxidant capacity (denoted here as *Y*) uses the weighed sum of thiolate-specific basicities, where the weights are the *χ* relative abundances of the thiolate-bearing microspecies. For cysteine for example:(10)$$Y_{\mathrm{CysSH}}=\chi_{a}\cdot \log k^{S}+\chi_{b}\cdot \log k_{N}^{S}+\chi_{d}\cdot \log k_{O}^{S}+\chi_{f}\cdot \log k_{\mathrm{NO}}^{S}$$

Note that all the four log*k* values bear the S superscript, indicating the important and exclusive role of thiolate basicities, as the only currently available, measurable parameters related to antioxidant capacity. The juxtaposed *χ* factors set the pH range where the respective a, b, d, and f microspecies occur abundantly enough to be the predominant reducing agent. The reason why the weighed sum is used instead of the weighed mean is because this calculation method reflects the nature of antioxidant capacity decreasing to 0 at very low pH, where the effective concentration of the reactive thiolate-bearing species also approaches 0. The *Y* values of selected thiolate-containing compounds are shown in Figure 5.

### 3.4. Designing for Optimal Antioxidant Capacity

Based on the *Y* values of antioxidant capacity defined above, one can calculate the optimal thiolate log*k* value which affords the maximal antioxidant capacity at a given pH. For a range of different log*k* values, *Y* is calculated (with Equation (10) using a monoprotic model, i.e., the thiolate moiety is assumed to have no interaction with any other basic moiety in the chemical space. This simulation was done at every pH unit and the results are depicted in Figure 6. It can be read from the graph that optimal thiolate basicity for antioxidant capacity near-physiological conditions (pH near 7) would be around log*k* 6.

## 4. Discussion

Since several serious illnesses have been shown [5,6] to be associated with oxidative stress, a preventively active antioxidant agent is an urgent medical need. The elaboration of such a molecule, however, assumes a narrow trail. While an optimal antioxidant drug has to be able to reductively eliminate the harmful, reactive oxidizing agent, it must not alter the reducible units in biomolecules, such as the disulfides in natural peptides and proteins.

The most promising candidates that have the appropriate range and variability of redox potentials are the thiol–disulfide and selenol–diselenide systems in molecules whose side-chain groups can beneficially modulate the redox properties of the thiols and selenols. Antioxidant activities based on thiol–disulfide and selenol–diselenide conversions are qualified by two factors: The appropriate reducing strength of the actually working thiolate/selenolate moiety in the antioxidant molecule; and the pH range where the concentration of the thiolate/selenolate-containing microspecies are high enough to exert sufficient antioxidant activity.

The subtle, selective antioxidant strength and pH appropriation can only be sized if the site-specific redox potentials are known. These latter parameters have long been inaccessible for reasons that: Traditional electrochemical methods fail to work, since electrode surfaces get irreversibly poisoned by thiols and selenols [34]; and indirect molecular methods by any of the assays in Gulcin’s list might show at best the overall redox potential, not the site-specific one.

These unresolved difficulties could be recently overcome by discovering a close correlation between the basicity and redox potential of thiolates. The same is true for selenolates. The correlation between the thiolate (selenolate) basicity and oxidizability stems from the electron density on the sulfur (selenium) atom. The higher the electron density, the higher the basicity, and the easier it is to release an electron from (i.e., to oxidize) the sulfur (selenium) atom. High electron density is caused by nearby electron-sending group(s) in the molecule, such as aliphatic moieties. Decreased electron density can be due to electron-withdrawing moieties, which are typically oxygen- or nitrogen-containing groups, such as carboxyl, carboxylate, ammonium, amino, hydroxyl, phenolic, nitro, etc. However, relatively high electron density on the sulfur (selenium) atom causes an adverse effect on the pH range where the thiolate (selenolate) group can act as antioxidants. The enhanced electron density is concomitant with higher basicity and elevated pH range where the hydrogen ion of the thiol (selenol) group dissociates, a precondition of the subsequent oxidation of the thiolate (selenolate) site. These subtle relationships specify the antioxidant capability of a particular compound. *Y* value that comprises the products of species-specific thiolate basicities and the relative concentration of the species in question, shows that typical thiolates (cysteine, cysteamine, glutathione) gain large *Y* values (i.e., considerable antioxidant capacities) at pH = 8 and above, since all these log*k*^S^ values are 8 and above (Figure 5). This means a high pH to lose the thiol proton, and high electron density and ease to release a sulfur electron. On the other hand, ovothiol A has unusually low log*k*^S^ values, since all the surrounding groups (ammonium, imidazolium, carboxyl) withdraw electrons from the sulfur. Its thiol therefore releases its proton at a pH of around 2, exerting and thus reducing activities at such a low pH. This reducing power is, however, low, because the electron density at this thiolate is also low. At higher pH, where all the adjacent groups deprotonate (ammonium to amino, imidazolium to imidazole, carboxyl to carboxylate) and thus the electron-withdrawing effects lessen and the thiolate electron density increases, resulting in a larger *Y* value and enhanced reducing capability. Nevertheless, this reducing capability still remains below that of cysteamine, cysteine, and glutathione, since the thiolate basicities of the latter are higher in each case. This means that the thiol form of ovothiol A is a reducing agent even at low pH, with weak or moderate reducing power. It also means practically that ovothiol A is hard to keep in its thiol (reduced) form.

The three-dimensional plot of the antioxidant capacity (Figure 6) shows iso-reductive surfaces for *Y* values as a function of both the log*k*^S^ thiolate basicity and the solution pH, to find the optimal, selective compound to eliminate the harmful oxidizing agent, while keeping intact the reducible units of essential biomolecules, under any (patho)physiological pH circumstances. This model is in agreement with the works of Nagy [4], whereby it is established that an optimal thiolate basicity regarding antioxidant activity is around log*k*^S^ 6.

A comparison of antioxidant capacities (*Y*) calculated in this work with experimental data reveals that a more consistent determination of antioxidant capacities is needed for thiol compounds. Regarding ovothiol, a study [35] reports superoxide scavenging assay results for an ovothiol analogue and glutathione, in which near pH 7 the mercaptoimidazole derivative is found superior, in agreement with the proposed antioxidant capacity in Figure 5. A more detailed experimental analysis is needed to further confirm the validity of the theoretical calculations in this work, in which the antioxidant capacity of the discussed thiols is measured using reliable assays at different pH points.

## 5. Conclusions

The development of an effective antioxidant agent is a critical medical need due to the association between oxidative stress and various serious illnesses. The ideal antioxidant must selectively neutralize harmful oxidizing agents without disrupting vital biomolecules. Thiolate–disulfide and selenolate–diselenide systems are promising candidates due to their adaptable redox potential. The antioxidant effectiveness of these systems is influenced by the reducing strength of the active moiety and the pH range where it is active. Recent advancements have linked thiolate/selenolate basicity with their redox potential, providing a way to fine-tune antioxidant properties. The proposed measures of antioxidant capacity in this work suggest that compounds with a thiolate basicity around six are optimal for balancing selectivity and effectiveness when considering cytosolic pH environments. As with all antioxidant capacity measures, the proposed methods only highlight one aspect of thiolate/selenolate antioxidant activity. The reducing potential of these compounds estimated from their basicity values determined in near physiological conditions, should still be handled with scrutiny when considering the complex biological environment and multifaceted antioxidant pathways of these biological compounds.

## Figures and Tables

**Figure 1 antioxidants-13-01053-f001:**
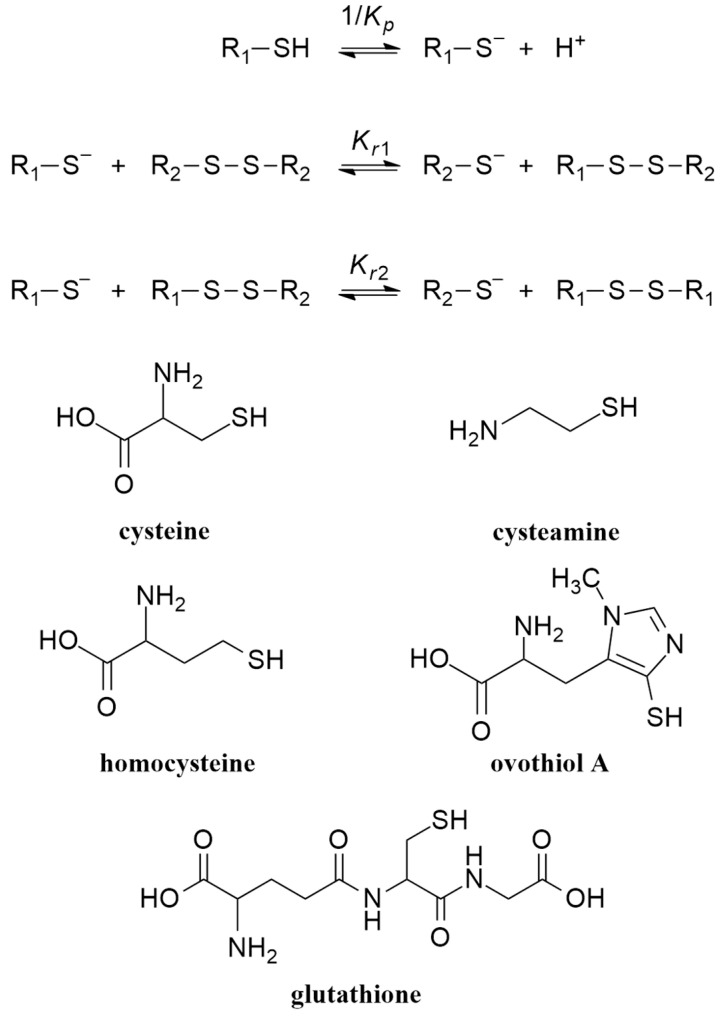
Structure of the most important thiol compounds; deprotonation of thiols, and the two-step redox exchange equilibria between thiolate and disulfide species. Note that often the net reaction of the above two-step redox reactions is depicted.

**Figure 2 antioxidants-13-01053-f002:**
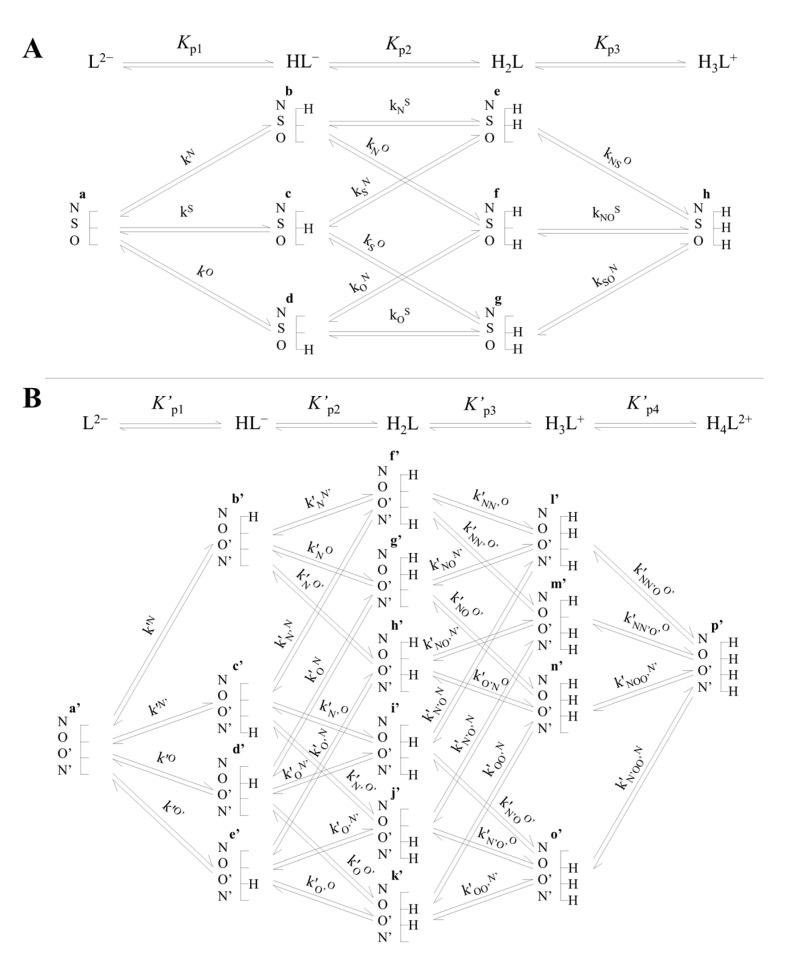
The species-specific protonation equilibrium schemes (microspeciation) of cysteine (**A**) and cystine (**B**). The schematic structure of the compounds shows the microspecies with its basic sites where N, S, and O labels denote the amino, thiolate, and carboxylate groups, respectively. Top lines represent the macroscopic protonation steps. Cysteine microspecies are labeled with one-letter symbols (a, b, c, …), while cystine microspecies are depicted with one-letter followed by a prime (a′, b′, c′, …).

**Figure 3 antioxidants-13-01053-f003:**
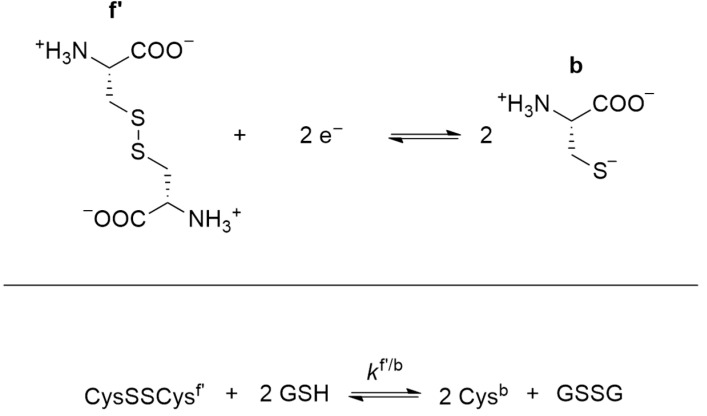
The redox half-reaction between microspecies (f′) of cystine and microspecies (b) of cysteine (**top**). The above redox couple can be in equilibrium with the GSSG and GSH redox forms of glutathione. The net reaction of the latter two-step redox exchange is shown (**bottom**).

**Figure 4 antioxidants-13-01053-f004:**
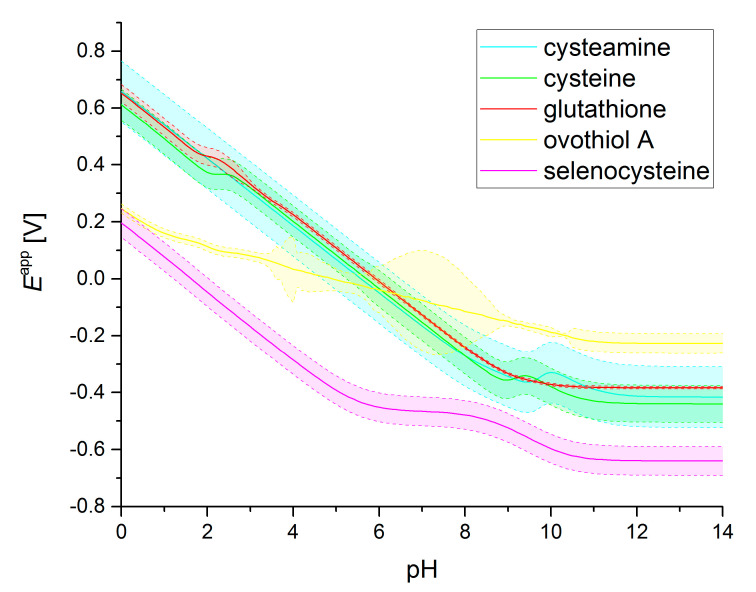
The calculated pH-dependent apparent redox potential curves of select thiolate- and selenolate-containing compounds (with 95% confidence bands). The apparent redox potential values were obtained using Equations (6) and (9) and analogous ones, by setting the concentration of all compounds to 1 mol/L (e.g., for $E_{\mathrm{CysSSCys}/\mathrm{CysSH}}^{\mathrm{app}}$, $\;{\left[\mathrm{CysSH}\right]}_{T}={\left[\mathrm{CysSSCys}\right]}_{T}=1$). Note that if the concentrations are changed, different curves might result. Note that a spline smoother was applied to the curve of ovothiol A to avoid a non-monotonous section between pH 4 and 10. Small deviations of non-monotonicity in the other curves are due to the varying E values from different microspecies.

**Figure 5 antioxidants-13-01053-f005:**
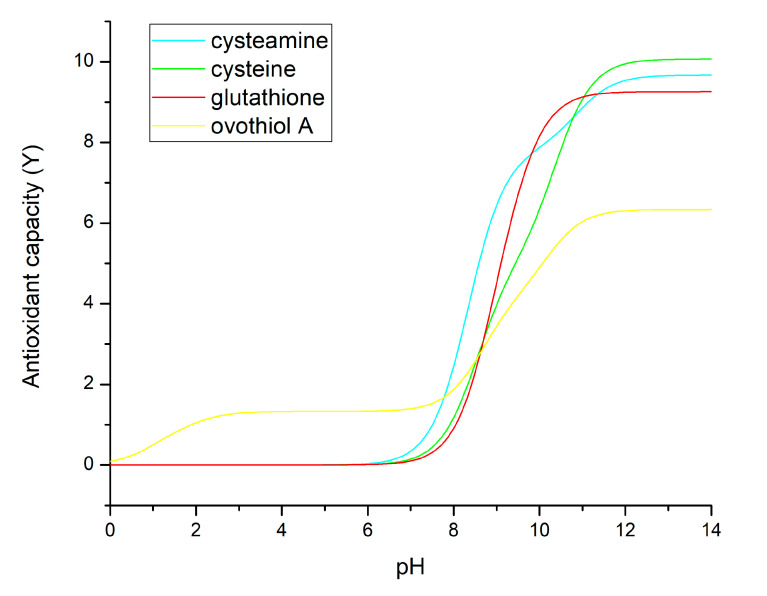
Antioxidant capacities (*Y*) calculated directly from thiolate-specific basicities and the concomitant relative abundance of the thiolate-bearing species as a function of pH. The standard error of *Y* is on the order of 0.1.

**Figure 6 antioxidants-13-01053-f006:**
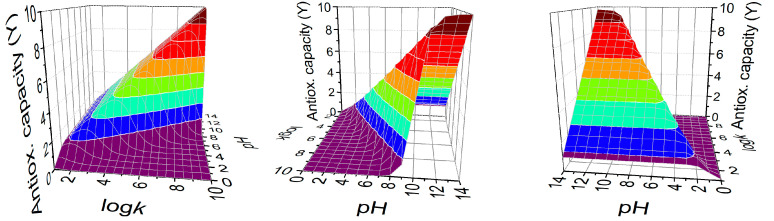
The 3D plot of the antioxidant capacity (*Y*) shows which thiolate logk is optimal for antioxidant strength at each pH. Color contour lines indicate constant *Y* values.

**Table 1 antioxidants-13-01053-t001:** The predicted *E*° values of protein thiolate moieties (reported with the lower and upper ends of 95% confidence intervals) calculated from the experimentally determined thiolate-specific logk values.

Protein Name	PDB ID	Res ID	log*k*	*E*° [V] (95% CI)
Alpha-1-antitrypsin	1QLP	232	6.86	−0.247 (−0.253; −0.241)
Hydroperoxide reductase c	4MA9	46	5.94	−0.191 (−0.198; −0.185)
Human DJ-1	1P5F	106	5.4	−0.159 (−0.166; −0.152)
Creatine kinase	1I0E	283	5.6	−0.171 (−0.177; −0.164)
O6-Alkylguanine-DNA alkyltransferase	1EH6	145	5.3	−0.153 (−0.160; −0.145)
Tyrosine phosphatase 1b	2HNP	215	5.57	−0.169 (−0.176; −0.162)
Papaya protease omega	1PPO	25	2.88	−0.006 (−0.017; 0.005)
Yersinia protein tyrosine phosphatase	1YPT	403	4.67	−0.114 (−0.123; −0.106)
Cathepsin B	1THE	29	3.6	−0.050 (−0.060; −0.040)
DsbA protein	1DSB	30	3.5	−0.044 (−0.054; −0.033)
Thioredoxin	2TRX	32	7.1	−0.261 (−0.267; −0.256)
	2TRX	35	9.9	−0.431 (−0.438; −0.423)
Disulfide isomerase	1MEK	36	4.5	−0.104 (−0.113; −0.096)
Ubiquitin conjugating enzyme	1JBB	87	11.1	−0.504 (−0.513; −0.494)
Ubiquitin-conjugating enzyme 2B	1JAS	88	10.2	−0.449 (−0.457; −0.441)
Human thioredoxin	1ERT	32	6.3	−0.213 (−0.219; −0.207)
Glutaredoxin	1EGO	11	5.5	−0.165 (−0.172; −0.158)
	1EGO	14	10.5	−0.467 (−0.476; −0.459)
Acyl-coenzyme A binding protein	1NTI	86	9.9	−0.431 (−0.438; −0.423)

## Data Availability

The raw data supporting the conclusions of this article will be made available by the authors on request.

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
