# Peer review of "Prediction of Antioxidant Capacity of Thiolate–Disulfide Systems Using Species-Specific Basicity Values"

_antioxidants, 2024, doi:10.3390/antiox13091053_

Round 1
Reviewer 1 Report
A manuscript entitled “Prediction of antioxidant capacity of thiolate-disulfide systems using species-specific basicity values” by Tamás Pálla et al., was submitted to Antioxidants for possible consideration as a research article. In this study, authors utilized the named correlation between thiolate basicity and oxidizing ability, to quantify capacities and pH-dependences of thiol-disulfide antioxidant systems, as a useful tool to search and assess proper molecules against oxidative stresses.
Some minor issues should be addressed before the full acceptance of this manuscript to be processed even further.
1) I checked up the Tab. 1, thioredoxin, with one PDB entry number but two items, why? The name of the protein was not specific.
2) I am afraid that the provided list of the proteins was inadequate. To fully describe the important prediction of antioxidant capacity of thiolate-disulfide systems, the authors should list as much more proteins as possible, giving adequate values from different species and with detailed basicities.
3) Why Sec was shown in Fig. 4 but lack in Figure 5?
4) Regarding the Fig. 5, the Y axis was not directly noted.
5) Why not showing the Fig. 6 from different angels?
6) As for Figs. 4, 5, and 6, all of them were theoretical calculations, right? Did authors try the experimental verifications on some pH ranges or select points?
7) Where was the major conclusion of the whole manuscript?
Minor revision can be suggested on the present manuscript.
A manuscript entitled “Prediction of antioxidant capacity of thiolate-disulfide systems using species-specific basicity values” by Tamás Pálla et al., was submitted to Antioxidants for possible consideration as a research article. In this study, authors utilized the named correlation between thiolate basicity and oxidizing ability, to quantify capacities and pH-dependences of thiol-disulfide antioxidant systems, as a useful tool to search and assess proper molecules against oxidative stresses.
Some minor issues should be addressed before the full acceptance of this manuscript to be processed even further.
1) I checked up the Tab. 1, thioredoxin, with one PDB entry number but two items, why? The name of the protein was not specific.
2) I am afraid that the provided list of the proteins was inadequate. To fully describe the important prediction of antioxidant capacity of thiolate-disulfide systems, the authors should list as much more proteins as possible, giving adequate values from different species and with detailed basicities.
3) Why Sec was shown in Fig. 4 but lack in Figure 5?
4) Regarding the Fig. 5, the Y axis was not directly noted.
5) Why not showing the Fig. 6 from different angels?
6) As for Figs. 4, 5, and 6, all of them were theoretical calculations, right? Did authors try the experimental verifications on some pH ranges or select points?
7) Where was the major conclusion of the whole manuscript?
Minor revision can be suggested on the present manuscript.
Author Response
Response to Reviewer 1:
A manuscript entitled “Prediction of antioxidant capacity of thiolate-disulfide systems using species-specific basicity values” by Tamás Pálla et al., was submitted to Antioxidants for possible consideration as a research article. In this study, authors utilized the named correlation between thiolate basicity and oxidizing ability, to quantify capacities and pH-dependences of thiol-disulfide antioxidant systems, as a useful tool to search and assess proper molecules against oxidative stresses.
Some minor issues should be addressed before the full acceptance of this manuscript to be processed even further.
Comments 1: I checked up the Tab. 1, thioredoxin, with one PDB entry number but two items, why? The name of the protein was not specific.
Response 1: The PDB ID “2TRX” refers to THIOREDOXIN FROM ESCHERICHIA COLI: https://www.rcsb.org/structure/2TRX. In this protein 2 cysteine residues can be found in positions 32 and 35, hence the 2 separate rows for the same protein.
Comments 2: I am afraid that the provided list of the proteins was inadequate. To fully describe the important prediction of antioxidant capacity of thiolate-disulfide systems, the authors should list as much more proteins as possible, giving adequate values from different species and with detailed basicities.
Response 2: Unfortunately, the complete list of known thiolate basicities in proteins in literature is such a short list as is included in Table 1 of the manuscript.
Comments 3: Why Sec was shown in Fig. 4 but lack in Figure 5?
Response 3: As was detailed in the manuscript, the antioxidant capacity measure shown in Figure 5 is capable of comparing only thiolate compounds or only selenolate compounds. With this metric the comparison of thiolate species with selenolate species is not meaningful, therefore selenocysteine cannot be included in Figure 5.
Comments 4: Regarding the Fig. 5, the Y axis was not directly noted.
Response 4: The Y axis is labelled ‘Y’ - the symbol chosen for this antioxidant capacity as elaborated in Section 3.3, this is indeed confusing, therefore the label of the Y axis in Figure 5 was changed to ‘Antioxidant capacity (Y)’.
Comments 5: Why not showing the Fig. 6 from different angels?
Response 5: Figure 6 is replaced showing the graph from 3 multiple angles.
Comments 6: As for Figs. 4, 5, and 6, all of them were theoretical calculations, right? Did authors try the experimental verifications on some pH ranges or select points?
Response 6: The data shown in Figures 4-6 are from theoretical calculations using experimental data. The experimental verification of these values (standard redox potential, antioxidant capacity (Y) calculated from thiolate basicities) is not possible, since these values are not directly observable. A comparison can be made in the future to compare the calculated antioxidant capacity values to the result of different antioxidant assays (e.g. CUPRAC or peroxynitrite scavenging assay).
Comments 7: Where was the major conclusion of the whole manuscript?
Response 7: The following Conclusions were added to the manuscript “The development of an effective antioxidant agent is a critical medical need due to the association between oxidative stress and various serious illnesses. The ideal antioxidant must selectively neutralize harmful oxidizing agents without disrupting vital biomolecules. Thiolate-disulfide and selenolate-diselenide systems are promising candidates due to their adaptable redox potentials. The antioxidant effectiveness of these systems is influenced by the reducing strength of the active moiety and the pH range where it is active. Recent advancements have linked thiolate/selenolate basicity with their redox potential, providing a way to fine-tune antioxidant properties. The proposed measures of antioxidant capacity in this work suggests that compounds with a thiolate basicity around 6 are optimal for balancing selectivity and effectiveness when considering cytosolic pH environments. As with all antioxidant capacity measures, the proposed methods only highlight one aspect of thiolate/selenolate antioxidant activity. The reducing potential of these compounds estimated from their basicity values determined in near physiological conditions, should still be handled with scrutiny when considering the complex biological environment and multifaceted antioxidant pathways of these biological compounds.”.
Minor revision can be suggested on the present manuscript.
Reviewer 2 Report
The present manuscript deals with predicting the antioxidant capacity of thiolate-disulfide systems using species-specific basicity values. In my opinion, the idea of this work is not so original and the manuscript should be significantly improved in the context of the literature. The introduction and discussion lack key references.
Major comments
1. In the abstract, the authors state “This study utilizes the recently discovered correlation between thiolate basicities and oxidizabilities, to quantify capacities and pH-dependences of thiol-disulfide antioxidant systems, as a tool to find adequate molecules against oxidative stress”. I think that this sentence should be rephrased. I mean, the correlation between thiolate basicities and oxidizabilities as well as between thiols acidity and reductive potential is well known in chemistry. This is not a recent discovery. Maybe the authors mean that it is possible to predict antioxidant capacity from basicities values of thiolates. The cited work only refers to a mathematical calculation for the prediction.
2. In the introduction, the authors should remove the list of all the antioxidant assays available, citing only some examples. On the contrary, for the methods useful for thiols: CUPRAC and peroxynitrite scavenger assay, they should add the relative references.
3. In the introduction, the authors should explain to the readers the players of their study and the relative biological roles with appropriate references. I mean glutathione, cystine, ovothiols, I don’t think everyone knows them and their importance for such studies, especially for ovothiols. I think it may be useful to show also the chemical structure of these thiols, in order to help the reader to understand the reasons of their antioxidant properties.
4. In the discussion, the authors state “These latter parameters have long been inaccessible for reasons that: Traditional electrochemical methods fail to work since electrode surfaces get irreversibly poisoned by thiols and selenols.” This citation needs references.
5. The authors should compare the results obtained from their calculations with the redox potentials previously measured for such compounds. See for example the following papers for comparison and references: Free Radic Biol Med. 1995, 18:679-85; Nat. Prod. Rep. 2018, 35, 1241–1250; Antioxidants 2021, 10, 1470; ACS Omega 2022, 7, 31813–31821.
6. In the conclusions, the authors should discuss whether their calculations fit with the antioxidant capacity previously reported for such compounds, clearly showing which are the compounds with the most antioxidant capacity. The authors should also explain the limits of their calculation compared to the complex biological environment of the cell.
7. In the methods, the authors should cite the database used for collecting data.
Minors
-Add space after figure legends.
-Line 261 262 remove carriage return.
Author Response
Response to Reviewer 2:
The present manuscript deals with predicting the antioxidant capacity of thiolate-disulfide systems using species-specific basicity values. In my opinion, the idea of this work is not so original and the manuscript should be significantly improved in the context of the literature. The introduction and discussion lack key references.
Comments 1: In the abstract, the authors state “This study utilizes the recently discovered correlation between thiolate basicities and oxidizabilities, to quantify capacities and pH-dependences of thiol-disulfide antioxidant systems, as a tool to find adequate molecules against oxidative stress”. I think that this sentence should be rephrased. I mean, the correlation between thiolate basicities and oxidizabilities as well as between thiols acidity and reductive potential is well known in chemistry. This is not a recent discovery. Maybe the authors mean that it is possible to predict antioxidant capacity from basicities values of thiolates. The cited work only refers to a mathematical calculation for the prediction.
Response 1: We agree with the Reviewer and the sentence in the Abstract was changed to “This study utilizes the known correlation between thiolate basicities and oxidizabilities, to quantify antioxidant or reducing capacities and pH-dependences of thiol-disulfide antioxidant systems, as a tool to find adequate molecules against oxidative stress”.
Comments 2: In the introduction, the authors should remove the list of all the antioxidant assays available, citing only some examples. On the contrary, for the methods useful for thiols: CUPRAC and peroxynitrite scavenger assay, they should add the relative references.
Response 2: We agree with the Reviewer, and instead of the list of all the antioxidant assays the following was added:
“The most important in vitro antioxidant assays, summarized by Gulcin [9], are for example oxygen radical absorbance capacity (ORAC), total radical trapping antioxidant parameter (TRAP), trolox equivalence antioxidant capacity assay (TEAC), and many others. However, of the many antioxidant assays available, only cupric ions (Cu2+) reducing antioxidant power assay (CUPRAC) and peroxynitrite scavenging assays are capable of measuring the antioxidant activity of thiolates. The CUPRAC assay, developed by Apak’s group [DOI: 10.1080/09637480600798132], measures antioxidant capacity by reducing Cu²⁺ to Cu⁺ in the presence of neocuproine, forming a Cu⁺-neocuproine complex with a peak absorption at 450 nm. This method is cost-effective, stable, and suitable for a wide range of antioxidants, including both hydrophilic and lipophilic types. It operates at a pH close to physiological conditions (7.0), making it more applicable to potential in vivo antioxidant reactions, and is particularly effective for measuring thiol-type antioxidants. Peroxynitrite (ONOO⁻) is a potent and short-lived oxidant that contributes to neurodegeneration and various diseases like Alzheimer's and cancer. It is formed by the reaction of nitric oxide with superoxide and can diffuse across cell membranes, causing cellular damage through lipid peroxidation, protein oxidation, and DNA strand breakage. Due to its stability relative to other free radicals, peroxynitrite can induce significant tissue injury, yet there are no endogenous enzymes to inactivate it, highlighting the need for specific scavengers. The oxidation of dihydroxyrhodamine by peroxynitrite can be measured using fluorescence spectrophotometry, providing a method to assess its scavenging activity [DOI: 10.1016/j.jsps.2012.05.002].”.
Comments 3: In the introduction, the authors should explain to the readers the players of their study and the relative biological roles with appropriate references. I mean glutathione, cystine, ovothiols, I don’t think everyone knows them and their importance for such studies, especially for ovothiols. I think it may be useful to show also the chemical structure of these thiols, in order to help the reader to understand the reasons of their antioxidant properties.
Response 3: We agree with the Reviewer, and the following elaboration of the mentioned thiols was added:
“
Cysteine, cysteamine, homocysteine, and glutathione are sulfur-containing molecules with critical biological roles [https://doi.org/10.1016/j.freeradbiomed.2019.05.035]. Cysteine is essential for protein synthesis and acts as a precursor to glutathione, a key antioxidant. Cysteamine serves as a protective agent against radiation and oxidative stress, and is used in treating cystinosis. Homocysteine is an intermediate in methionine metabolism, with elevated levels linked to cardiovascular disease. Glutathione is a powerful antioxidant that protects cells from oxidative damage and detoxifies harmful substances, playing a crucial role in maintaining cellular redox balance. Ovothiol A, B, and C are natural 4-mercaptohistidine derivatives, varying in methylation at the amino site, found in marine organisms, particularly in sea urchin eggs. Ovothiol is a functional group dense molecule with multiple protonation states, making it one of the most chemically diverse small biomolecules. It is zwitterionic at physiological pH, features strong imidazole-thiolate interactions, and has unique thiolate protonation constants, resulting in its sulfur atom predominantly existing in an anionic form across the pH scale. This versatility enhances its antioxidant capabilities without altering its molecular structure [doi: 10.1007/s00216-014-7631-0]. Ovothiol has a unique ability to scavenge reactive oxygen species (ROS) and protect against DNA, protein, and lipid damage [doi: 10.3390/antiox10091470].
”.
Also Figure 1 was updated to include the structural formulae.
Comments 4: In the discussion, the authors state “These latter parameters have long been inaccessible for reasons that: Traditional electrochemical methods fail to work since electrode surfaces get irreversibly poisoned by thiols and selenols.” This citation needs references.
Response 4: The following citation was added https://doi.org/10.1021/ac000716c.
Comments 5: The authors should compare the results obtained from their calculations with the redox potentials previously measured for such compounds. See for example the following papers for comparison and references: Free Radic Biol Med. 1995, 18:679-85; Nat. Prod. Rep. 2018, 35, 1241–1250; Antioxidants 2021, 10, 1470; ACS Omega 2022, 7, 31813–31821.
Response 5: The authors thank the Reviewer for the relevant citations pertaining to the antioxidant capacity of ovothiol A. However, the second reference contains no data on measured antioxidant capacity, the third reference contains detailed kinetic studies, however not in relation to other thiol antioxidants, and the fourth reference introduces Gibbs Reaction Energies, which are again a direct derivative of basicity values (Table 2.) In the first reference the antioxidant capacity of d 1-methyl-5-ethyl-4-mercaptoimidazole (MEMI) is measured in comparison with that of glutathione using 2 various methods (trolox and superoxide assays). MEMI is an ovothiol analogue and some conclusions can be made based on these results. The main obstacle in comparing our theoretical results with experimental data is that these thiol-containing compounds have not been measured with the same assay (e.g. CUPRAC) in a consistent manner. The antioxidant capacity of homocysteine for example was reported using Rosup assay (doi: 10.3390/antiox12010202). Perhaps in the future a more direct comparison can be made between theoretical and experimental data. The following was added to the Discussion section:
“Comparison of antioxidant capacities (Y) calculated in this work with experimental data reveals that a more consistent determination of antioxidant capacities is needed for thiol-compounds. Regarding ovothiol, a study [https://doi.org/10.1016/0891-5849(94)00186-N] reports superoxide scavenging assay results for an ovothiol analogue and glutathione, in which near pH 7 the mercaptoimidazole derivative is found superior, in agreement with the proposed antioxidant capacity in Figure 5. A more detailed experimental analysis is needed to further confirm the validity of the theoretical calculations in this work, in which the antioxidant capacity of the discussed thiols is measured using reliable assays at different pH points.”.
Comments 6: In the conclusions, the authors should discuss whether their calculations fit with the antioxidant capacity previously reported for such compounds, clearly showing which are the compounds with the most antioxidant capacity. The authors should also explain the limits of their calculation compared to the complex biological environment of the cell.
Response 6: The comparison of the theoretical calculations was made with experimental data as detailed in Response 5. The limitation of the work was also added “As with all antioxidant capacity measures, the proposed methods only highlight one aspect of thiolate/selenolate antioxidant activity. The reducing potential of these compounds estimated from their basicity values determined in near physiological conditions, should still be handled with scrutiny when considering the complex biological environment and multifaceted antioxidant pathways of these biological compounds.”.
Comments 7: In the methods, the authors should cite the database used for collecting data.
Response 7: The citation of doi: 10.1093/database/baz024 was added to the appropriate part of the Methods section.
Minors
Comments 8: Add space after figure legends.
Response 8: The necessary formatting was done.
Comments 9: Line 261 262 remove carriage return.
Response 9: The necessary formatting was done.
Round 2
Reviewer 2 Report
Even though the authors improved the introduction and the paper's discussion, much of the stated information lacks references and the results are not significantly discussed in the existing literature.
lines 69-70 "Homocysteine is an intermediate in methionine metabolism, with elevated levels linked to cardiovascular disease" add reference;
lines 71-72 "Glutathione is a powerful antioxidant that protects cells from oxidative damage and detoxifies harmful substances, playing a crucial role in maintaining cellular redox balance". Add reference.
lines 73-74 "Ovothiol A, B, and C are natural 4-mercaptohistidine derivatives, varying in methylation at the amino site, found in marine organisms, particularly in sea urchin eggs." Add appropriate reference.
lines 75-78 "Ovothiol is a functional group dense molecule with multiple protonation states, making it one of the most chemically diverse small biomolecules. It is zwitterionic at physiological pH, features strong imidazole-thiolate interactions, and has unique thiolate protonation constants, resulting in its sulfur atom predominantly existing in an anionic form across the pH scale" Add appropriate reference.
In Table 1, the authors report pKa and redox potential of cysteine-containing proteins, however along the text the authors never report the pka and E0 previously reported for cysteine, glutathione, and ovothiol, which are the main actors of the manuscript. In NPR 2018, in paragraph 3 prof. Seebeck reviewed the pka and E0 for all such compounds, with appropriate references. This information is extremely relevant to the present manuscript and should be taken into account for the discussion about the antioxidant capacities of such compounds.
Author Response
Response to Reviewer (round 2):
Comments 1: lines 69-70 "Homocysteine is an intermediate in methionine metabolism, with elevated levels linked to cardiovascular disease" add reference;
Response 1: The following reference was added: doi: 10.1186/1475-2891-14-6
Comments 2: lines 71-72 "Glutathione is a powerful antioxidant that protects cells from oxidative damage and detoxifies harmful substances, playing a crucial role in maintaining cellular redox balance". Add reference.
Response 2: The following reference was added: doi: 10.3390/molecules20058742
Comments 3: lines 73-74 "Ovothiol A, B, and C are natural 4-mercaptohistidine derivatives, varying in methylation at the amino site, found in marine organisms, particularly in sea urchin eggs." Add appropriate reference.
Response 3: The following reference was added: https://doi.org/10.1038/srep21506
Comments 4: lines 75-78 "Ovothiol is a functional group dense molecule with multiple protonation states, making it one of the most chemically diverse small biomolecules. It is zwitterionic at physiological pH, features strong imidazole-thiolate interactions, and has unique thiolate protonation constants, resulting in its sulfur atom predominantly existing in an anionic form across the pH scale" Add appropriate reference.
Response 4: The following reference was added: DOI: 10.1007/s00216-014-7631-0
Comments 5: In Table 1, the authors report pKa and redox potential of cysteine-containing proteins, however along the text the authors never report the pka and E0 previously reported for cysteine, glutathione, and ovothiol, which are the main actors of the manuscript. In NPR 2018, in paragraph 3 prof. Seebeck reviewed the pka and E0 for all such compounds, with appropriate references. This information is extremely relevant to the present manuscript and should be taken into account for the discussion about the antioxidant capacities of such compounds.
Response 5: The pKa and redox potentials discussed by Prof. Seebeck are indeed of great relevance, however these cited values are macroscopic protonation constants and pH-dependent redox potential values. In our work the emphasis is on species-specific parameters. The following paragraph was added to section 3.1. Relationship between acid-base and redox characteristics.
“The acid-base and redox parameters referenced from work [16: doi:10.1038/srep37596] are species-specific ones, characterizing protonation or oxidation transitions on a submolecular level pertaining to a particular protonation state, as opposed to the more frequently used macroscopic pKa and apparent redox potential values. Previously reported [DOI: 10.1039/c8np00045j, https://doi.org/10.1002/cbic.201600228, https://doi.org/10.1021/jo00027a023] macroscopic parameters for cysteine (thiol pKa = 8.4, redox potential = −0.22 V) and glutathione (thiol pKa = 8.7, redox potential = −0.26 V) are distinct but in agreement with the species-specific values; i.e. if pH 7 is considered the species-specific acid-base and redox parameters can be calculated to an apparent redox potential in agreement with previous findings.”